# Expression of *PnSS* Promotes Squalene and Oleanolic Acid (OA) Accumulation in *Aralia elata* via Methyl Jasmonate (MeJA) Induction

**DOI:** 10.3390/genes14061132

**Published:** 2023-05-23

**Authors:** Honghao Xu, Wenxue Dai, Meiling Xia, Wenhua Guo, Yue Zhao, Shunjie Zhang, Wa Gao, Xiangling You

**Affiliations:** 1Key Laboratory of Saline-Alkali Vegetation Ecology Restoration, Ministry of Education, Northeast Forestry University, Harbin 150040, China; xhhnefu@163.com (H.X.); 15049163983@163.com (W.D.); xmlnefu@163.com (M.X.); gwh0631@126.com (W.G.); zhaoyue4694@163.com (Y.Z.); 2Medical Resources Research Center, Mudanjiang Branch of Heilongjiang Academy of Forestry Sciences, Mudanjiang 157011, China; zhangshunjie2200@163.com; 3Application of Nuclear Technology, Heilongjiang Institute of Atomic Energy, Harbin 150081, China; gw_6019@163.com

**Keywords:** *Aralia elata*, genetic transformation, heterologous expression, squalene, oleanolic acid, MeJA treatment

## Abstract

*Aralia elata* is an important herb due to the abundance of pentacyclic triterpenoid saponins whose important precursors are squalene and OA. Here, we found that MeJA treatment promoted both precursors accumulation, especially the latter, in transgenic *A. elata*, overexpressing a squalene synthase gene from *Panax notoginseng*(*PnSS*). In this study, *Rhizobium*-mediated transformation was used to express the *PnSS* gene. Gene expression analysis and high-performance liquid chromatography (HPLC) were used to identify the effect of MeJA on squalene and OA accumulation. The *PnSS* gene was isolated and expressed in *A. elata*. Transgenic lines showed a very high expression of the *PnSS* gene and farnesyl diphosphate synthase gene (*AeFPS*) and a slightly higher squalene content than the wild-type, but endogenous squalene synthase (*AeSS*), squalene epoxidase (*AeSE*), and β-amyrin synthase (*Aeβ-AS*) gene were decreased as well as OA content. Following one day of MeJA treatment, the expression levels of *PeSS*, *AeSS*, and *AeSE* genes increased significantly. On day 3, the maximum content of both products reached 17.34 and 0.70 mg·g^−1,^ which increased 1.39- and 4.90-fold than in the same lines without treatment. Transgenic lines expressing *PnSS* gene had a limited capability to promote squalene and OA accumulation. MeJA strongly activated their biosynthesis pathways, leading to enhance yield.

## 1. Introduction

*A. elata* (Miq.) Seem., which belongs to the Araliaceae family, is an important saponin-yielding woody plant [1]. This species is particularly rich in triterpenoid saponins that are utilized for preventive and curative healthcare, TCM therapy, and medical research [2]. The major constituents of triterpenoid saponins in *A. elata* are oleananes and dammaranes [3,4].

In most plant species, triterpenoid saponin biosynthesis occurs via the mevalonate (MVA) pathway, although the cross-talk with plastidial methyl erythritol phosphate (MEP) pathway can occur under certain circumstances [5]. Many studies have demonstrated that triterpenoids are synthesized in *A. elata* primarily via the MVA pathway [1,4,6,7], which comprises the six main successive enzymatic reactions from condensation via farnesyl pyrophosphate synthase (FPS), squalene synthase (SS), squalene epoxidase (SE), and ß-amyrin synthase (ß-AS) to glycosylation via glucosyltransferase (UGTs) (Figure 1) [2]. Among them, SS catalyzes the second committed step. When the *PgSS1* gene was transformed into *Eleutherococcus senticosus*, its activity in transgenic somatic embryos was upregulated by approximately 3-fold compared with that in the WT, and the levels of seven ciwujianosides increased by between 2- and 2.5-fold [8]. In the leaves of *Hedera helix*, the high expression of *HhSS* was consistent with hederacoside accumulation [9]. Gao et al. identified a positive correlation between the expression levels of the *PpSS1* gene and the accumulation of steroidal saponins in *Paris polyphylla* rhizomes, indicating that the *PpSS1* gene plays an important role in steroidal saponin biosynthesis [10].

Numerous studies have indicated that the expression levels of *SS* genes were up-regulated by MeJA functioning as an abiotic elicitor of bioactive substance biosynthesis in plants. Zhang reported that MeJA induced an 8-fold increase in the expression level of the *TwSS* gene in *Tripterygium wilfordii* compared with the control group [11]. In *Bupleurum falcatum*, MeJA treatment increased BfSS1 activity and protein levels. Moreover, following MeJA treatment, a higher content of the bioactive substances squalene, total phytosterol including sitosterol, campesterol, stigmasterol, and saikosaponins were detected in adventitious roots of transgenic *B. falcatum* [12]. Expression of the SS protein and its encoding gene (*MsSQS*) expression in the model legume, *Medicago sativa*, which is rich in triterpene saponins, was rapidly increased in stem, leaf, and root following MeJA treatment, suggesting that MeJA treatment induced upregulation of the *MsSQS* gene and contributed to the accumulation of total saponins [13]. In *Araliaceae*, MeJA treatment also increased the *SS* gene transcript levels. In *Panax ginseng*, the transcript levels of the *PgSS1* and *SE* genes were significantly upregulated by 24 h after MeJA treatment [14]. Similarly, in *P. notoginseng*, the transcript levels of the *SS* (*PnSS*) gene and *SE* increased 24 h after MeJA treatment [15].

In this study, to investigate the biosynthesis of squalene and OA response to *SS* expression and MeJA application in *A. elata*, a *PnSS* was inserted into the somatic embryos of *A. elata*. The expression levels of key enzyme genes relating to triterpenoid saponins biosynthesis, squalene, and OA production capabilities of these transgenic plants were in contrast with the wild-type.

## 2. Materials and Methods

### 2.1. Isolation of PnSS Gene and Construction of Plant Expression Vectors

Somatic *A. elata* embryos were cultured using a previously described method [6]. Total RNA of *P. notoginseng* seedling was extracted using the Plant RNA Isolation Mini kit (BioTeKe Corporation, Wuxi, Jiangsu, China). A cDNA library was then established using the Prime Script^TM^ RT reagent kit with gDNA Eraser (Takara Bio Inc., Dalian, Liaoning, China) according to the manufacturer’s instructions. The *PnSS*-F and *PnSS*-R primers were designed according to the sequence of the reported *PnSS* gene (GenBank: DQ457054) and incorporated *Xba* I and *Sac* I restriction sites, respectively (Table 1). The PCR was performed using the following conditions: 3 min pre-denaturation at 94 °C, followed by 30 cycles of 30 s denaturation at 94 °C, 30 s annealing at 54 °C, and 2 min extension at 72 °C, with a 7 min full extension, and finally stored at 16 °C. Electrophoresed by 1% agarose gel, the PCR product, was recovered with an Omega E.Z.N.A^TM^ Gel Extraction Kit (Omega Bio-tek, Guangzhou, China). The recovered product was cloned into *pEASY*^®^-Blunt Cloning Vector (TransGen Biotech, Beijing, China). The recombinant plasmids were extracted with an Omega E.Z.N.A^TM^ Plasmid Mini Kit I (Omega Bio-tek, Guangzhou, China) and sequenced for confirmation. Then, the recombinant plasmids and pROKII plasmids were digested by *Xba* I and *Sac* I. The digestion product was electrophoresed, respectively. The target fragment of *PnSS* gene and linearized pROKII plasmids were recovered and ligated with a DNA Ligation Kit Ver.2.1 (Takara Bio Inc., Dalian, Liaoning, China). The pROKII-*PnSS* were identified by sequencing and transformed into *Agrobacterium Rhizogenesis* strains GV3101.

### 2.2. Genetic Transformation and PCR Detection

#### 2.2.1. Transformation of PnSS Gene

Plantlets from somatic embryogenesis of *A. elata* were obtained in tissue culture bottles and root explants were transformed previously described [6]. The transformed roots explants were then cultured using selection medium (MS basal medium supplemented with 0.8 mg·L^−1^ 2,4-D and 3% (*w·v*^−1^) sucrose, 40 mg·L^−1^ kanamycin, 300 mg·L^−1^ cefotaxime and 200 mg·L^−1^ Timentin, and solidified with 0.58% (*w*·*v*^−1^) agar). Calluses induced in explants were then used to generate independent callus lines by the method as previously described [6], which were transferred to the differentiation medium (SH basal medium supplemented with 3.0 mg·L^−1^ IBA and 3% (*w·v*^−1^) sucrose, and solidified with 0.58% (*w*·*v*^−1^) agar) for 20 days to induce somatic embryos.

#### 2.2.2. PCR Detection

Expression of *PnSS* gene in transformed callus and somatic embryos was validated with the T5 Direct PCR kit (Tsingke Biotechnology, Beijing, China) according to the manufacturer’s instructions. The PCR was performed in 50 µL reaction volume using the *PnSS*-F and *PnSS*-R primers (Table 1) under the following conditions: 3 min pre-denaturation at 98 °C, followed by 35 cycles of 10 s denaturation at 94 °C, 10 s annealing at 58 °C, and 15 s extension at 72 °C, with a 7 min full extension, and finally stored at 4 °C. A sample (5 µL) of the reaction mixture was checked by 1% (*w·v*^−1^) agarose gel electrophoresis and then visualized by GelStains (TransGen Biotech, Beijing, China) staining viewed under a UV transilluminator.

### 2.3. Expression of Key Enzymes

#### 2.3.1. Total RNA Extraction and cDNA Reverse-Transcript Synthesis

Total RNA of *A. elata* was isolated using Universal Plant Total RNA Extraction kit (spin-column) (BioTeKe Corporation, Wuxi, Jiangsu, China) according to the manufacturer’s instructions. The total RNA was used as a template to synthesize first-strand cDNA using the Prime Script^TM^ RT reagent kit with gDNA Eraser (Takara Bio Inc., Dalian, Liaoning, China) according to the manufacturer’s instructions.

#### 2.3.2. Gene Expression Analysis by Quantitative Real Time PCR (qRT-PCR)

The expression of five genes (*AeFPS*, *AeSS*, *AeSE*, *Aeβ-AS*, and *PnSS*) involved in triterpenoid saponins biosynthesis pathways were analyzed in somatic embryos by qRT-PCR using the TransStart^®^ Tip Green qPCR SuperMix Kit (TransGen Biotech, Beijing, China) and primers designed using the PrimerQuest™ (https://sg.idtdna.com/PrimerQuest/Home/Index, accessed on 12 October 2022) (Table 2). The reactions were performed in triplicate under the following conditions: 95 °C for 5 min; 40 cycles of 95 °C for 5 s, 60 °C for 34 s, and 72 °C for 30 s. The *GAPDH* gene (GenBank accession number: JQ183068.1) was used as a reference gene to normalize the data for quantification of the relative expression of the five genes using the 2^−ΔΔCT^ method.

### 2.4. Measurement of Squalene and OA

The transgenic somatic embryos confirmed by PCR were dried and the contents of squalene and OA were analyzed by the high-performance liquid chromatography (HPLC) system, Waters 1525-2707-2489, (Waters) and the XTerraMS C18 column (4.6 mm × 250 mm, 5 μm, Waters). The squalene content was measured using a previously described method with modifications [16]. Briefly, the mobile phase consisted of acetonitrile and ultrapure water at a flow rate of 1.0 mL·min^−1^, and the detection was performed at a wavelength of 210 nm at 25 °C. The OA content was measured as described previously [17]. Briefly, the mobile phase consisted of acetonitrile/water (9:1, *v·v*^−1^) at a flow rate of 1.0 mL·min^−1^, and the detection was performed at a wavelength of 210 nm at room temperature. The squalene and OA contents measured by HPLC were analyzed using the Origin 2021_v9.8.0.200_x64 software.

### 2.5. MeJA Treatment

Transgenic somatic embryos (1.5 g) were suspended in a liquid differentiation medium containing 200 μM MeJA for different periods (1, 3, and 5 d) to analyze its effect on the biosynthesis of the squalene and OA; three biological repeats were prepared.

### 2.6. Statistical Analysis

Data were presented as mean ± standard deviation. All statistical analysis was performed using IBM SPSS Statistics 26. Differences between groups were evaluated using Duncan’s multiple comparisons test and *p* < 0.05 was set as the threshold for statistical significance.

## 3. Results

### 3.1. Vector Construction and Transformation of PnSS Gene into A. elata

The total RNA was extracted from *P. notoginseng* (Appendix A) and reverse transcribed into cDNA which was verified with the *GAPDH* gene (Appendix A). The ORF of *PnSS* gene was amplified using *PnSS*-F and *PnSS*-R (Appendix A). The result of sequencing was following the ORF sequence of the *PnSS* gene (GenBank: DQ186630.1). A schematic diagram of the plant expression vector overexpressing *PnSS* gene under the control of CaMV 35S promoter is shown in Figure 2.

Roots explants were grown onto the cultivation medium (MS basal medium supplemented with 0.8 mg·L^−1^ 2,4-D and solidified with 0.58% (*w·v*^−1^) agar) (Figure 3A). After infection, the explants were transferred to the selection medium (Figure 3B). After 4 weeks, calluses appeared on the roots and were propagated (Figure 3C). The somatic embryos were then induced from the kanamycin-resistant callus lines (Figure 3D). Finally, 31 kanamycin-resistant callus lines and 22 kanamycin-resistant somatic embryos were obtained, and the presence of *PnSS* gene was confirmed by PCR.

The presence of the *PnSS* ORF in transgenic callus lines (CLs) and transgenic somatic embryo lines (SLs) was confirmed by agarose gel electrophoresis. A distinct band of approximately 1400 base pairs corresponding to the size of the *PnSS* ORF was detected in six transgenic somatic embryo lines (SL 6, 9, 16, 18, 20, and 21), but not in the blank control without template DNA, thus confirming that the PCR was free from contamination (Figure 4).

### 3.2. Key Enzyme Gene Expression Analysis

As shown in Figure 5, expression levels of *PnSS* gene had variable effects on the expression of other key genes (*AeFPS*, *AeSS*, *AeSE*, and *Aeβ-AS*) encoding enzymes in the triterpenoid saponin biosynthesis pathway in different SLs (Figure 5). *AeSS* expression in SLs was significantly lower than that in wild-type (WT), indicating that the expression of the *PnSS* gene inhibited the expression of *AeSS*. Expression of *AeFPS*, which is upstream of *AeSS*, was upregulated in SLs compared to the WT. In contrast, the expression levels of *AeSE* and *Aeβ-AS*, which are downstream of *AeSS*, were similar to that of the WT, while the expression levels of *AeFPS* were upregulated.

### 3.3. Measurement of the Squalene and OA Contents

To confirm the effect of *PnSS* overexpression on bioactive substance biosynthesis in *A. elata*, the squalene and OA contents in transgenic somatic embryo lines were detected by HPLC, with retention times at 28.609 min and 7.996 min, respectively (Figure 6A–F), and quantified according to a squalene and OA standard curves established according to the regression equations (y = 2.948x − 1.068, R^2^ = 0.999) and (y = 30.06x + 0.010, R^2^ = 0.999), respectively. The squalene content in transgenic SLs was higher than that in the WT, with higher levels in SL 9 and 20 compared with the other lines (Figure 7A). In contrast, the OA content in transgenic SLs was lower than in the wild-type (Figure 7B), with lower levels in SL 16 and 18 compared with the other lines (Figure 7B).

### 3.4. MeJA Treatment

#### 3.4.1. Analysis of Key Enzyme Gene Expression Analysis under MeJA Treatment

The effect of MeJA treatment on the expression of key genes (*AeFPS*, *AeSS*, *AeSE*, and *Aeβ-AS*) encoding enzymes in the triterpenoid saponin biosynthesis pathway expression in transgenic somatic embryos was analyzed by qRT-PCR (Figure 8). Similar trends in the expression of *AeSE*, *AeSS*, and *PnSS* genes were observed in all transgenic somatic embryo lines. After treatment with MeJA for one day, the expression levels of all three genes were higher in the transgenic lines than those at the later time points, with the highest expression of *AeSE* in SL 20 and the highest expression of both *AeSS* and *PnSS* genes in SL 16 (Figure 8A–C). In SL 16, 18, and 20, *Aeβ-AS* expression increased after one-day MeJA treatment, especially in SL 16, and it was 4.58-fold than its control group (Figure 8E,F). In SL 6 and 9, *AeFPS* expression was decreased at one-day post-treatment, and the inhibition was progressively enhanced at the later time points. *AeFPS* expression reached a minimum at day 3 post-treatment, but was significantly increased on day 5. In SL 16, 18, 20, and 21, *AeFPS* expression peaked at one-day post-treatment and decreased rapidly to 3-day, followed by an increase in expression on day 5 (Figure 8D).

#### 3.4.2. Measurement of the Squalene and OA Contents under MeJA Treatment

The effect of MeJA on the accumulation of squalene and OA contents in somatic embryos was analyzed by HPLC (in Figure 9). The amounts of squalene and OA in all six transgenic lines were higher than that in the WT (Figure 9A,B), reaching the highest levels at day 3, and followed by a decrease in the contents at later time points. Furthermore, the squalene and OA contents in SLs treated with MeJA for 3 days increased relative to the levels in SLs without treatment (Figure 9C,D). In SL16 treated with MeJA, the squalene and OA contents increased by 1.39 and 4.90-fold, respectively, compared with this line detected without treatment.

## 4. Discussion

Because both squalene and OA, as natural medicines, are secondary metabolites with great economical value, enhanced production of squalene and/or OA is an effective strategy to gain a larger benefit. The technique of genetic transformation is an attractive method to modify the biosynthesis of secondary metabolites via overexpression of key enzyme genes. On the other hand, the application of elicitors such as MeJA is one of the common biotechnological approaches to improve the biosynthesis of secondary metabolites via regulating expression levels of one or more genes. In *Senna obtusifolia* transgenic hairy roots with overexpression of *PgSS1* gene, MeJA treatment provided a significant effect on the biosynthesis of Betulinic Acid [18]. In *B. falcatum* with overexpression of *BfSS1* gene, MeJA treatment elevated the levels of Squalene, phytosterols, and saikosaponins [12]. In this study, we analyzed the gene-to-metabolite network of the five key enzyme genes (*PnSS*, *AeSS*, *AeSE*, *AeFPS*, and *Aeβ-AS*) and the two secondary metabolites (squalene and OA), which were affected by overexpression of *PnSS* gene in transgenic plants. On this basis, MeJA abiotic elicitor was applied to transgenic plants to further promote squalene and OA accumulation. 

In transgenic lines, the overexpression of the *PnSS* gene was accompanied by up-regulation of *AeFPS* gene expression (Figure 5), which was unexpected. In *Saccharomyces cerevisiae*, *FPS* gene independently participates in the biosynthesis of the MVA pathway products [19]. In other words, there is no connection between the expression of *FPS* and *SS* genes in *S. cerevisiae*. In *Centella asiatica* hairy roots, there was no correlation between the expression of *SS* and *FPS* genes [20]. However, to improve triterpenoid biosynthesis, a fusion protein, as a valuable tool including very high activities of both *FPS* and *SS*, was constructed and expressed in *Escherichia coli*, which suggests that further research about the positive correlation between expression of *PnSS* and *AeFPS* gene has great potential [21].

Overexpression of *PnSS* gene resulted in suppression of *Aeβ-AS* gene expression (Figure 5). A similar result was found in *B. falcatum* overexpressing *BfSS1* gene, in which mutual inhibition might exist between the end product of BfSS1 and β-AS [12]. In *S. cerevisiae*, the squalene overaccumulation proved to be an essential limiting factor for the biosynthesis of β-amyrin because of the inhibition effect of squalene on β-AS from *Glycyrrhiza glabra* [22]. However, contrary to the negative relativity between the expression of *SS* and *β-AS* genes in *A. elata* and *B. falcatum*, in *P. ginseng*, the expression level of *β-AS* gene was strongly upregulated by overexpression of *PgSS1* gene [23]. It indicates that the downregulation of *Aeβ-AS* gene expression might be attributed to the similar regulation system for the biosynthesis pathway of triterpene in *A. elata* and *B. falcatum*, but is different from in *P. ginseng* because triterpenoid saponins in *A. elata* and *B. falcatum* are mainly oleanane-type, while in *P. ginseng* they are mainly of the dammarane-type [6,23,24].

Additionally, overexpression of *PnSS* gene led to a decrease in the expression levels of *AeSS* and *AeSE* genes (Figure 5). In *P. notoginseng* and *A. elata*, *PnSS* and *AeSS* play the same role in the biosynthesis pathway of triterpenoid saponins, which combine two molecules of farnesyl diphosphate (FPP) to form one molecule of squalene. The suppression of *AeSS* gene expression might be attributed to the inhibition caused by functional identity. In *Taraxacum koksaghyz*, the downregulation of *TkSE1* gene expression produced by RNA interference led to a significant decrease in *TkSS1* gene expression [25]. In *P. ginseng*, the expression of *PgSS1* and *SE* genes was also coregulated [23]. These results support a positive correlation between the expression of *SS* and *SE* gene in the triterpenoid biosynthesis pathway. Therefore, it is assumed that suppression of *AeSS* gene expression resulted in the downregulation of *AeSE* gene expression.

An enhancement of squalene accumulation via overexpression of *SS* gene has been reported in *Synechocystis* sp. PCC 6803 [26]. However, in *Chlamydomonas reinhardtii*, the squalene overaccumulation was not due to overexpression of *CrSS* gene but partial knockdown of *CrSE* gene [27]. In transgenic lines overexpressing *PnSS* gene, although *AeSS* gene expression was suppressed, squalene production was enhanced (Figure 7A). It might be mainly attributed to the decrease in *AeSE* gene expression instead of overexpression of *PnSS* gene. According to the results of gene expression analysis, the decline of another important product OA was due to the clear suppression of expression of *AeSE* and *Aeβ-AS* genes (Figure 6 and Figure 7B).

Generally, MeJA is considered an efficient elicitor to regulate lots of physiological and biochemical reactions in plants such as the expression of defense genes followed by the accumulation of secondary metabolites [28]. Some reports suggested that MeJA application may further intensify certain characteristics which were not changed obviously [29,30]. The first and the last step of scopolamine biosynthesis were catalyzed by Putrescine N-methyltransferase (PMT) and Hyoscyamine-6-hydroxylase (H6H), respectively. Zhang transformed a *PMT* gene of *Nicotiana tabacum* into *Hyoscyamus niger* hairy roots in which the yield of tropane alkaloids did not significantly increase although the PMT activity increase 5-fold as compared to wild-type. However, after MeJA application on transgenic plants, scopolamine accumulation was found. The results of reverse transcription PCR (RT-PCR) analysis showed that MeJA treatment enhanced the expression of *PMT* as well as *H6H* gene and then the downstream scopolamine biosynthesis pathway was activated [29]. A similar situation happened with another report. Zheng reported that the overexpression of the rice (E)-b-caryophyllene synthase gene (*OsTPS3*) in transgenic rice plants cannot result in the emission of this volatile metabolite in large quantities. However, MeJA treatment made it emit more. The results of qPCR analysis showed that MeJA not only stimulated *OsTPS3* but also *OsFPS* expression. The latter synthesized a large quantity of the substrates of (E)-b-caryophyllene, FDP (Farnesyl pyrophosphate), which led to these volatile sesquiterpenes emitted in large amounts following MeJA treatment [30]. In our study, the same results were obtained. When transgenic embryo lines overexpressing *PnSS* gene were exposed to MeJA (200 μM) on day 1, the expression levels of the *PnSS*, *AeSS*, and *AeSE* genes were strongly induced (Figure 8A–C). Additionally, SL 16 under treatment presented nearly 5-fold higher *Aeβ-AS* gene expression levels than SL 16 without treatment (Figure 8E,F). Furthermore, SL 16 under 3-day MeJA treatment presented 4.90-fold higher OA content than SL 16 (Figure 9D). These results indicate that MeJA, as a signaling molecule, effectively reversed the suppression of OA biosynthesis caused by overexpression of *PnSS* gene.

## 5. Conclusions

In this study, the overexpression of *PnSS* gene could enhance the squalene production but inhibit OA accumulation, which might be caused by the antagonism of squalene overaccumulation on the activities of Aeβ-AS. We, for the first time, demonstrate this inhibition could be completely relieved by MeJA treatment.

Because of the negative correlation between the expression of *AeSE* and *PnSS* genes, overexpression of both *PnSS* and *AeSE* genes might be another efficient strategy to promote OA biosynthesis in *A. elata*.

## Figures and Tables

**Figure 1 genes-14-01132-f001:**
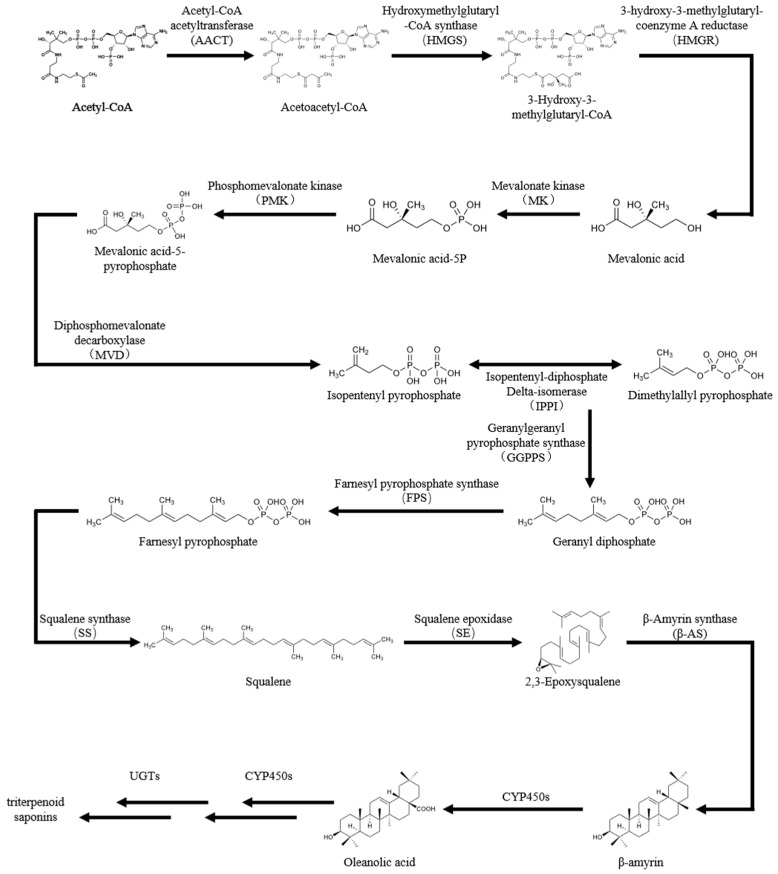
The biosynthesis of oleanane-type triterpenoid saponins via the MVA pathway.

**Figure 2 genes-14-01132-f002:**
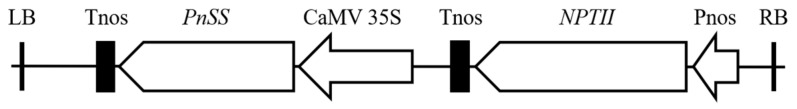
Schematic diagram of the *PnSS* ORF under the control of the CaMV 35S promoter, which was inserted into ProKII-Plant vector. *Tnos* denotes the terminator region of the nopaline synthase (*Nos*) gene from the Ti plasmid of *Rhizobium rhizogenes*. *Pnos* denotes the promoter region of the *Nos* gene. *NPTII* denotes the gene encoding neomycin phosphotransferase II, which confers resistance cells towards aminoglycoside antibiotics such as kanamycin by their inactivation via phosphorylation in prokaryotic and eukaryotic cells. LB denotes T-DNA left border. RB denotes T-DNA right border.

**Figure 3 genes-14-01132-f003:**
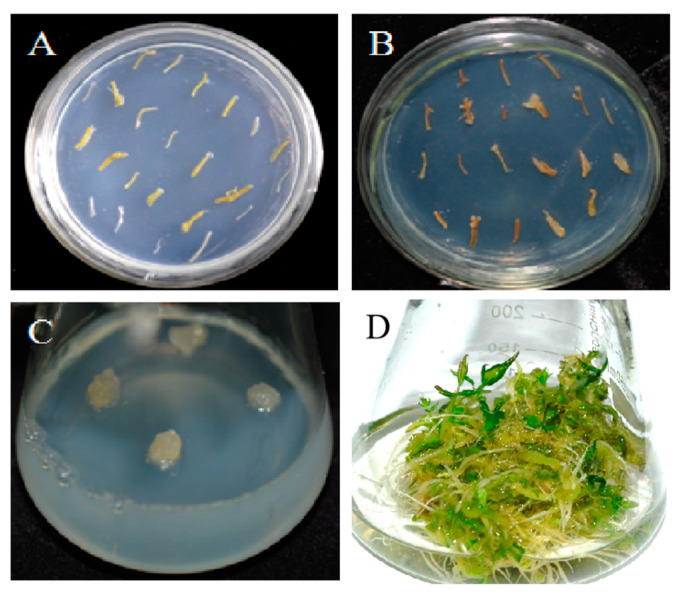
*Rhizobium*-mediated *PnSS* transformation of *A. elata* using adventitious roots removed from apices as explants. (**A**) Apices removed from adventitious roots grown on cultivation medium; (**B**) Explants after infection with *R. rhizogenes* on cultivation medium; (**C**) Calluses induced from explants on selection medium; (**D**) Differentiated transgenic calluses formed somatic embryos in *A. elata* on the differentiation medium.

**Figure 4 genes-14-01132-f004:**
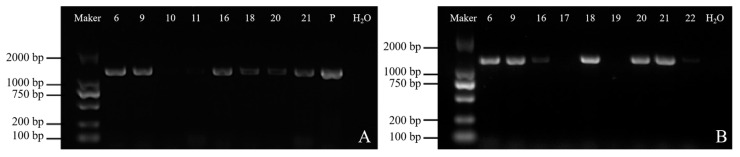
(**A**) PCR detection of CLs (CL 6, 9, 10, 11, 16, 18, 20, and 21) with primers specific to the *PnSS* gene; (**B**) PCR detection of SLs (B, SL 6, 9, and 16–22) with primers specific to the *PnSS* gene. (Marker) DNA 2 Kb marker. (P) Positive control (bacterial solution). (H_2_O) Blank control without template DNA.

**Figure 5 genes-14-01132-f005:**
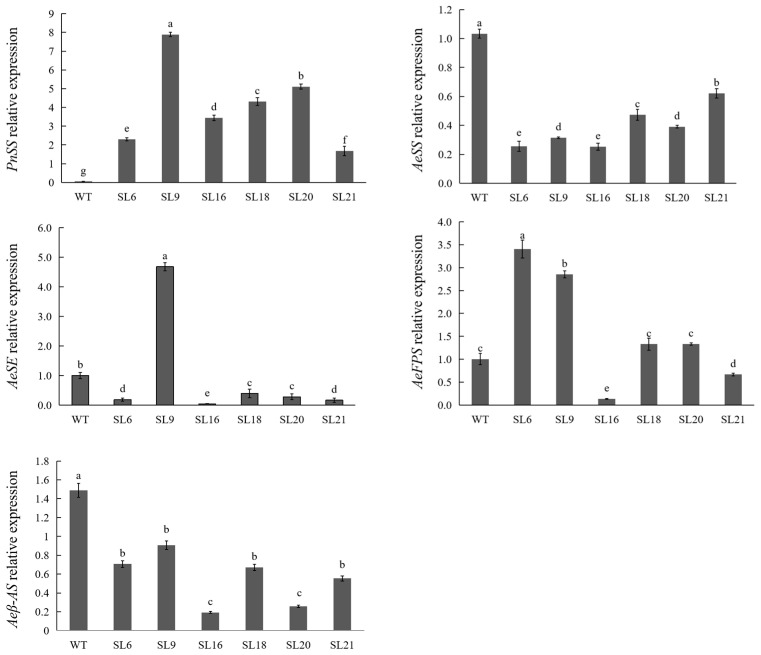
Analysis of the relative expression level of genes (*AeFPS*, *AeSS*, *AeSE*, *Aeβ-AS*, and *PnSS*) involved in triterpenoid saponin biosynthesis in wild-type (WT) and SL 6, 9, 16, 18, 20, and 21. Different lower-case letters indicate significant differences at *p* < 0.05.

**Figure 6 genes-14-01132-f006:**
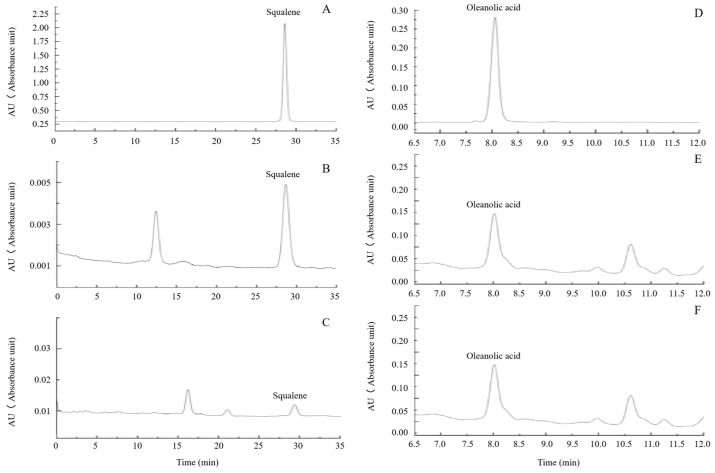
HPLC detection of squalene and OA. (**A**–**C**) HPLC of squalene standard, transgenic SL and WT, respectively; (**D**–**F**) HPLC of OA standard, transgenic SL and WT, respectively.

**Figure 7 genes-14-01132-f007:**
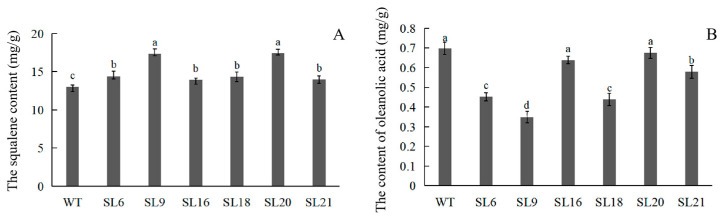
HPLC analysis of squalene (**A**) and OA (**B**) content in transgenic somatic embryos and WT. Transgenic somatic embryo lines: SL 6, 9, 16, 18, 20, and 21. Different lower-case letters indicate significant differences at *p* < 0.05.

**Figure 8 genes-14-01132-f008:**
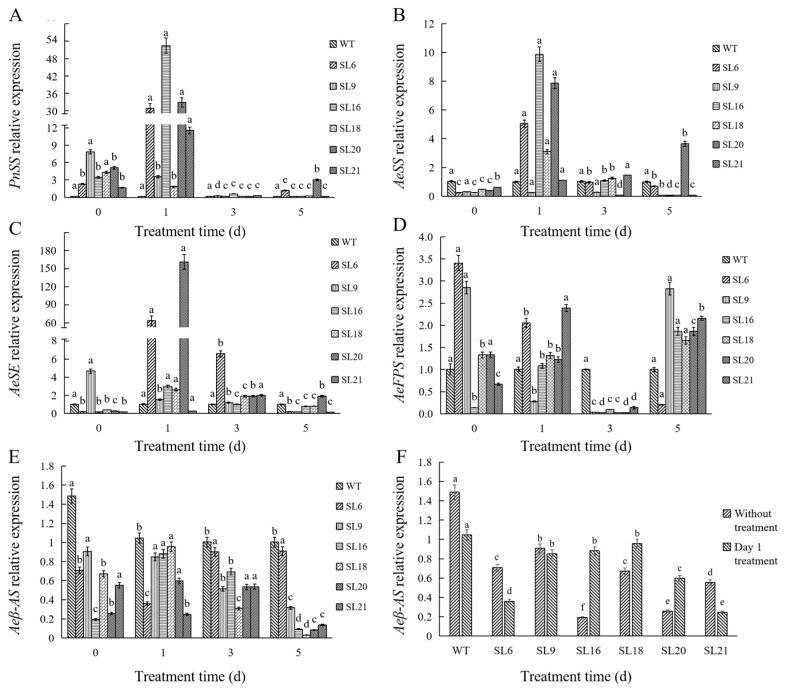
The relative expression levels of key genes (*PnSS*, *AeSS*, *AeSE*, *AeFPS*, and *Aeβ-AS*) involved in triterpenoid saponin biosynthesis in somatic embryos under MeJA treatment (0, 1, 3, and 5 d). In (**A**–**E**), different lower-case letters indicate significant differences among the same line on different treatment days at *p* < 0.05. In (**F**), different lower-case letters indicated significant differences among the six lines under the same condition at *p* < 0.05.

**Figure 9 genes-14-01132-f009:**
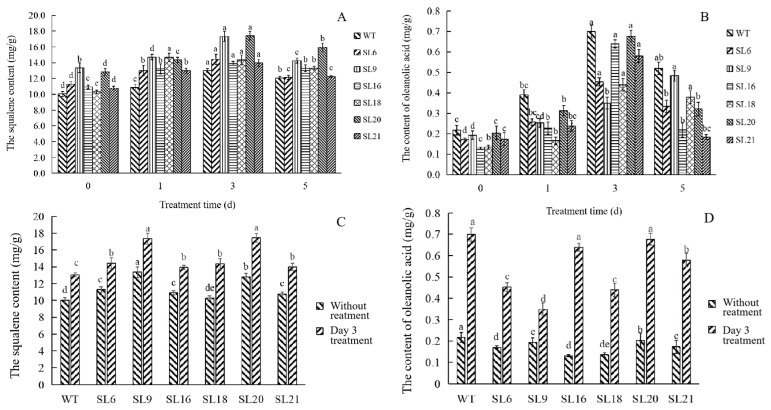
HPLC analysis of squalene (**A**) and OA (**B**) content in transgenic somatic embryos and WT under MeJA treatment for different hours (0, 1, 3, and 5 d). HPLC analysis of squalene (**C**) and OA (**C**) content in transgenic somatic embryos and WT under MeJA treatment for 3 days. In (**A**,**B**), different lower-case letters indicate significant differences among the same line on different treatment days at *p* < 0.05. In (**C**,**D**), different lower-case letters indicate significant differences among the six lines under the same condition at *p* < 0.05.

**Table 1 genes-14-01132-t001:** Sequences of the PCR amplification.

Primer Names	Primer Sequences (5′→3′)
*PnSS*-F	ATCTCTAGAGAGATGGGAAGTTTGGGGGCAATT
*PnSS*-R	ATCGAGCTCTCACTGTTTTTTCGGTAGTAGG

**Table 2 genes-14-01132-t002:** Sequences of the real time primers.

Gene Names	Primer Sequences (5′→3′)
*GAPDH* (JQ183068.1)	GGGAAAGTGCTACCTGCATTA
	CCACAAAGTCAGTGGAGACTAC
*AeFPS* (HM219226.1)	CCAGAGGTGATTGGGAAGATTG
	TGCTCTCATACTCGGCAAATAC
*AeSS* (GU354313.1)	GTGGAGACAGTGGGTGATTATG
	ACATGCGTGACTTTGGTATCT
*AeSE* (GU354314.1)	CCGGGATCTTCTTAGACCTTTAC
	TCCTCCGAGGCTCAGATAAT
*Aeβ-AS* (HM219225)	CTTCCTATGCACCCAGCTAAA
	CCCAGAGCAGGTCTTGTATTT
*PnSS* (DQ186630.1)	CCGGACGATTTCTATCCGTTAT
	CAGTGTCAAGTGCTCGAAGA

## Data Availability

The data presented in this study are available on request from the corresponding author.

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
