# Peer review of "Expression of PnSS Promotes Squalene and Oleanolic Acid (OA) Accumulation in Aralia elata via Methyl Jasmonate (MeJA) Induction"

_genes, 2023, doi:10.3390/genes14061132_

Round 1

Reviewer 1 Report

The manuscript is interesting for the Readers involved in the trials of enhancement of triterpenoid production, it contains some valuable results but it is badly prepared and written in rather awkward style.

The sentences like “ In this study, the OA content in the transgenic SL16 under MeJA treatment presented 4.90-fold higher than its no treatment.” are really not acceptable in scientific report. Without rewriting the text in a correct English this manuscript should not be accepted.

Some other comments:

Line 68. The aim of the study is not clear. The Authors have written: “In this study, for the purpose of investigating the secondary metabolism response to SS expression and MeJA application in A. elata, a PnSS was inserted into somatic embryos of A. elata.” Please be more precise, obviously it is not possible to investigate the response of entire secondary metabolism, this study was rather dedicated to the special group of metabolites.

Line 35. “In most species, triterpenoid saponin biosynthesis occurs via two pathways: the mevalonate (MVA) pathway and methyl erythritol phosphate (MEP)”. Of course that the cross-talk between these two pathways exists, however, from this sentence one can draw the conclusion that both MVA and MEP pathways are involved equally in saponin biosynthesis, which is not really justified. Even the Authors of the cited reference (Liao et al. Biotechnol. Adv. 2016, 34, 697-713.) wrote in their review that “one of the main roles of the most important enzyme in the MVA pathway, HMGR, is triterpenoid biosynthesis”. Therefore, it would be better to write that “In most plant species, triterpenoid saponin biosynthesis occurs via the mevalonate (MVA) pathway, although the cross-talk with plastidial methyl erythritol phosphate (MEP) pathway can occur under certain circumstances [5]”.

Lines 21-22. The conclusions are not clear. “Transgenic lines  expressing PnSS had a limited capability to promote two substances accumulation.” Which two substances? It should be stated precisely.

Line 41. “And it has been extensively researched studied in previous studies.” (what is the purpose of this sentence? In plus, it is awkward and it contains the unnecessary repetitions)

Line 58. phytosterol and saikosaponin, both should be in plural (phytosterols and saikosaponins), or it should be precisely written which phytosterol (sitosterol, stigmasterol, other?).

Line 74. Isolation PnSS – it should be isolation of PnSS, there are plenty of such mistakes in the manuscript.

Some parts of Material and Methods were “copied and pasted” even without adjusting the type of the font.

Line 133. “then analyzed the contents of squalene and OA by the high-performance liquid chromatography (HPLC): - rather: and the contents of squalene and OA were analyzed by the high-performance liquid chromatography (HPLC).

Line 145. For the treatment with 200 μM MeJA – how this solution was prepared? MeJA is not dissolved in water.

Line 313. “the improvement of OA in some transgenic line was more than that in its no treatment”. It is really a pity that the Authors cannot write even the conclusions in more scientific style.

The comments of the quality of English language are included in the main review. Unfortunatly, the poor quality of some parts of the text decreases the overall quality of the manuscript, and it is a pity, since the results are quite interesting.

Author Response

Dear Reviewer 1:

We appreciate your careful consideration of our manuscript and your positive comments and suggestions. Accordingly, we have improved the English writing of the text, described the aim of this study in more detail, and made extensive corrections to the previous manuscript. The revised text is marked in red, please see the attachment. Detailed point-by-point responses are listed below.

Comment: 1. Line 68. The aim of the study is not clear. The Authors have written: “In this study, for the purpose of investigating the secondary metabolism response to SS expression and MeJA application in A. elata, a PnSS was inserted into somatic embryos of A. elata.” Please be more precise, obviously it is not possible to investigate the response of entire secondary metabolism, this study was rather dedicated to the special group of metabolites.

Response: We have now replaced ‘the secondary metabolism with ‘the biosynthesis of squalene and OA’ in the manuscript Introduction. (Line 78-79)

Comment: 2. Line 35. “In most species, triterpenoid saponin biosynthesis occurs via two pathways: the mevalonate (MVA) pathway and methyl erythritol phosphate (MEP)”. Of course that the cross-talk between these two pathways exists, however, from this sentence one can draw the conclusion that both MVA and MEP pathways are involved equally in saponin biosynthesis, which is not really justified. Even the Authors of the cited reference (Liao et al. Biotechnol. Adv. 2016, 34, 697-713.) wrote in their review that “one of the main roles of the most important enzyme in the MVA pathway, HMGR, is triterpenoid biosynthesis”. Therefore, it would be better to write that “In most plant species, triterpenoid saponin biosynthesis occurs via the mevalonate (MVA) pathway, although the cross-talk with plastidial methyl erythritol phosphate (MEP) pathway can occur under certain circumstances [5]”.

Response: We have replaced “one of the main roles of the most important enzyme in the MVA pathway, HMGR, is triterpenoid biosynthesis” with “In most plant species, triterpenoid saponin biosynthesis occurs via the mevalonate (MVA) pathway, although the cross-talk with plastidial methyl erythritol phosphate (MEP) pathway can occur under certain circumstances”. (Line 41-45)

Comment: 3. Lines 21-22. The conclusions are not clear. “Transgenic lines  expressing PnSS had a limited capability to promote two substances accumulation.” Which two substances? It should be stated precisely.

Response: We have replaced “two substances” with “squalene and OA”. (Line 27-28)

Comment: 4. Line 41. “And it has been extensively researched studied in previous studies.” (what is the purpose of this sentence? In plus, it is awkward and it contains the unnecessary repetitions)

Response: We have deleted this sentence. (Line 50)

Comment: 5. Line 58. phytosterol and saikosaponin, both should be in plural (phytosterols and saikosaponins), or it should be precisely written which phytosterol (sitosterol, stigmasterol, other?).

Response: We have revised this sentence and added more details in line 67-69.

Comment: 6. Line 74. Isolation PnSS – it should be isolation of PnSS, there are plenty of such mistakes in the manuscript.

Response: We have revised this sentence in line 84, and revised these mistakes in the text.

Comment: 7. Some parts of Material and Methods were “copied and pasted” even without adjusting the type of the font.

Response: We have unified the type of the font in parts of Material and Methods.

Comment: 8. Line 133. “then analyzed the contents of squalene and OA by the high-performance liquid chromatography (HPLC): - rather: and the contents of squalene and OA were analyzed by the high-performance liquid chromatography (HPLC).

Response: We have replaced “then analyzed the contents of squalene and OA by the high-performance liquid chromatography (HPLC)” with “and the contents of squalene and OA were analyzed by the high-performance liquid chromatography (HPLC)”. (Line 146-148)

Comment: 9. Line 145. For the treatment with 200 μM MeJA – how this solution was prepared? MeJA is not dissolved in water.

Response: We dilute 45.84 μL of the original MeJA solution with ethanol to 0.4 mL first. Then transfer it to a 10-mL volumetric flask, and dilute it with distilled water drop by drop to volume. The concentration of our MeJA mother liquor is 20 mM·L-1.

Comment: 10. Line 313. “the improvement of OA in some transgenic line was more than that in its no treatment”. It is really a pity that the Authors cannot write even the conclusions in more scientific style.

Response: We have deleted this sentence and revised Conclusion in line 405-411.

We hope that the revised manuscript is now acceptable for publication in the Genes.
    Thank you again.

Yours sincerely,
Honghao Xu

Reviewer 2 Report

The article submitted to me for review article entiltled ,, Expression of PnSS promotes squalene and oleanolic acid (OA) accumulation in Aralia elata via methyl Jasmonate (MeJA) induction’’ is original work. . The authors present enhance of MeJa on the accumulation of secondary metabolites in lines with overexpression of the PnSS gene in A. elata. The article is interesting and brings many important threads to deepen knowledge in plant metabolism and increase the production of secondary metabolites, and although MeJa is known to increase the production of active compounds, the transgenic nature and gene expression in A. elata is very interesting. The abstract and the Introduction are written correctly, but please pay attention to some stylistic errors. In addition, the aim of the work is quite laconic, which should clearly define what should be examined in the work? Please correct. The material and methods section is quite informative and does not raise any major objections. However, I would like to point out the nomenclature in the name of the bacteria. Currently, Agrobacterium rhizogenes functions as Rhizobium rhizogenes and this name should be used in the article, so please correct it. The results section is described correctly, while the discussion section is too general and suggests that the authors discuss the results more, especially since there are many articles that have not been presented by the authors e.g. doi: 10.3390/molecules26206208. Besides, the discussion, in my opinion, seems to be interrupted and unfinished, there is no clear conclusion- please rephrase.

 Minor editing of English language required

Author Response

Dear Reviewer 2,

Thank you for your comments concerning our article. We have revised our previous draft extensively according to your suggestions. We have improved the English writing of the text, and replaced ‘Agrobacterium rhizogenes’ with ‘Rhizobium rhizogenes’. We have rewritten Discussion and Conclusion. All changes have been marked in red in the text, please see the attachment.

We hope that the revised manuscript is now acceptable for publication in the Genes.
    Thank you again.

Yours sincerely,
Honghao Xu

Reviewer 3 Report

The manuscript discusses a study that focused on six transgenic PnSS somatic embryo lines and they found that MeJA treatment promoted precursors accumulation in transgenic A. elata. Overall, the manuscript is interesting and the work and experiments are described clearly, however, some corrections need to consider in the revised manuscript.

1-     The manuscript has minor grammar and punctuation problems and needs to be checked for the English language by a native speaker;

2-     In Table 2, reference gene primers used for gene expression need to be added.

3-      Generally, it would be a good idea to re-write the conclusion section. In this section, some results are repeated without any future perspective for the results.

4-     In references all scientific names of the species need to be italicized.

The manuscript has minor grammar and punctuation problems and needs to be checked for the English language by a native speaker;

Author Response

Dear Reviewer 3,

We thank you very much for taking the time to review this manuscript. We have revised our previous draft extensively according to your suggestions. All changes are highlighted in red, please see the attachment. Detailed point-by-point responses are listed below.   

Comment: 1. The manuscript has minor grammar and punctuation problems and needs to be checked for the English language by a native speaker;

Response: We have made extensive modification to the English writing of our manuscript.

Comment: 2. In Table 2, reference gene primers used for gene expression need to be added.

Response: We have added the reference gene primers and their sequences in Table 2.

Comment: 3. Generally, it would be a good idea to re-write the conclusion section. In this section, some results are repeated without any future perspective for the results.

Response: We have rewritten the discussion and the conclusion section.

Comment: 4. In references all scientific names of the species need to be italicized.

Response: We have revised the font of all scientific names in all references.

We hope that the revised manuscript is now acceptable for publication in the Genes.
    Thank you again.

Yours sincerely,
Honghao Xu

Round 2

Reviewer 1 Report

The Authors has improved the manuscript according to my suggestions. I have no more comments.

Reviewer 2 Report

please correct some stylistic errors resulting from changes in the mnauscript.